# A Social Return on Investment Evaluation of the Pilot Social Prescribing EmotionMind Dynamic Coaching Programme to Improve Mental Wellbeing and Self-Confidence

**DOI:** 10.3390/ijerph191710658

**Published:** 2022-08-26

**Authors:** Abraham Makanjuola, Mary Lynch, Ned Hartfiel, Andrew Cuthbert, Hayley T. Wheeler, Rhiannon Tudor Edwards

**Affiliations:** 1Centre for Health Economics and Medicines Evaluation, Bangor University College of Health and Behavioural Sciences, Bangor LL57 2PZ, UK; 2Lanarkshire Campus, Hamilton International Technology Park, University of West Scotland, South Lanarkshire, Paisley G72 0LH, UK; 3School of Medicine Cardiff, Cardiff University College of Biomedical and Life Sciences, Cardiff CF14 4EP, UK; 4EmotionMind Dynamic, Hayley T Wheeler Ltd., Llanelli SA15 1BQ, UK

**Keywords:** social return on investment (SROI), social cost–benefit analysis, lifestyle coaching, mental health, wellbeing, social prescribing

## Abstract

The COVID-19 pandemic contributed to longer waiting lists for people seeking to access mental health services. The NHS Five Year Forward View encourages the development of empowerment-based social prescribing interventions to supplement existing mental health programmes. Based in South Wales, EmotionMind Dynamic (EMD) is a lifestyle coaching programme that supports individuals suffering from anxiety or depression. In this evaluation of lifestyle coaching, a mixed-method social return on investment (SROI) methodology was used to value quantitative and qualitative data from face-to-face and online participants. Data collection took place between June 2021 and January 2022. Participants included both self-referred clients and those referred from health services. Mental wellbeing data were collected at baseline and at the end of the programme using the short Warwick–Edinburgh Mental Wellbeing Scale (SWEMWBS) and the General Self-Efficacy Scale (GSES). Baseline and follow-up data were available for 15 face-to-face participants and 17 online clients. Wellbeing valuation quantified and valued outcomes from participants. Results indicated that for every GBP 1 invested, lifestyle coaching generated social values ranging from GBP 4.12–GBP 7.08 for face-to-face clients compared with GBP 2.37–GBP 3.35 for online participants. Overall, lifestyle coaching generated positive social value ratios for both face-to-face and online clients.

## 1. Introduction

### 1.1. Background

One in four adults in the United Kingdom (UK) is affected by poor mental health in their lifetime [1]. Poor mental health negatively impacts a person’s ability to cope with life and make informed [1] decisions. Mental ill health has a substantial effect on life expectancy and is a key cause of health inequalities. Evidence shows that people with severe and enduring mental health problems die on average 10 years earlier than the general population [2,3,4].

During the COVID-19 pandemic, a backlog in accessing mental health services emerged, with an estimated 215,000 adults in the U.K. unable to receive a referral to secondary mental health treatment [5]. Increasing rates of poor mental health are evidenced by the ongoing crisis of limited access to NHS services that support adults with mental health challenges. As of February 2022, the waiting list for specialised mental health treatment in the U.K. had increased to 1.6 million people, with another 8 million unable to access the list despite the service having been deemed potentially beneficial for them [6].

Due to increased wait times for NHS mental health services, there is a need for innovative face-to-face and online interventions that can support people suffering from anxiety or depression [5,7]. One promising intervention is lifestyle coaching, a systematic and structured approach to helping people make positive changes in their lives. However, an economic analysis of lifestyle coaching for mental health is currently lacking [8].

Lifestyle coaching helps clients to set and attain personal goals in order to enhance their mental wellbeing. The approaches of lifestyle coaching derive from the person-centred theories and practices of Carl Rogers, the founder of humanistic psychology. Lifestyle coaching emphasises goal setting and focusing on the future, which is similar to the person-centred counselling advocated by Rogers [9].

Lifestyle coaching takes a holistic approach in which clients work in partnership with coaches to find emotional balance and meaning by setting and reaching attainable goals. It is recognised that progression towards life goals is associated with an increase in wellbeing [10]. For clients who may not require a clinical intervention, lifestyle coaching has the potential to achieve similar outcomes to those of counselling and psychotherapy; it can, thereby, reduce waiting lists for mental health services [11].

### 1.2. The EmotionMind Dynamic Programme

Founded in 2016 by Hayley T. Wheeler, EmotionMind Dynamic (EMD) lifestyle coaching offers a non-clinical approach combining coaching, mentoring, counselling, teaching, and mindfulness. EMD is person-centred and focuses on self-empowerment, self-knowledge, and emotional processing. EMD helps clients to overcome mindset barriers, unlock limiting beliefs, and unlearn negative emotional programming. EMD provides a guided self-help methodology to facilitate the acquisition of life skills, such as self-reflective introspection, self-analysis, problem solving, goal setting, the reconstruction of old and new knowledge, and taking action.

Before the first COVID-19 lockdown in March 2020, EMD was delivered through a series of six face-to-face interactive coaching sessions (Figure 1). An EMD facilitator provided information that clients integrated into their previous knowledge, beliefs, and self-perception to develop new self-knowledge and self-awareness. Descriptions of the six EMD sessions are listed below (Figure 1):

Prior to COVID-19, the face-to-face format was used as a social prescribing referral option for 70 clients within the Llanelli primary care cluster, which is located in an area of South Wales with moderate overall multiple deprivation [12]. Due to a series of COVID-19 lockdowns, an online version of EMD was created to replicate the face-to-face experience. The online version included six learning sections, covering 31 units of guided self-help for mental wellbeing. The learning materials were the same for both face-to-face and online clients.

Although many social prescribing clients have a clinical diagnosis and are being treated for mental health challenges, a large number of clients can be referred to a range of local, non-clinical services to support their health and wellbeing. EMD was developed as a novel mental health and wellbeing intervention suitable for individuals experiencing mental wellbeing challenges, regardless of whether they had received a clinical diagnosis. EMD is particularly suitable for clients experiencing mental wellbeing challenges, particularly individuals who may fall outside statutory mental health service referral criteria; it is also suitable for discharged patients seeking ongoing self-development and personal growth.

There is a paucity of evidence on the potential benefits of non-clinical mental healthcare interventions that differ in approach from the more conventional psychological counselling currently offered by the NHS. The aim of this study was to provide an economic evaluation of EMD’s role in supporting people with mental health challenges. Currently, an economic analysis of lifestyle coaching for mental health is lacking [8].

The purpose of the pilot EMD SROI evaluation was to appraise the effectiveness of the programme in enhancing mental wellbeing and generate an associated social cost-benefit analysis. Secondary outcomes of the pilot study were the assessment of the acceptability of the EMD programme delivery approach alternatives, i.e., either face-to-face or online, and an estimation of the associated social value ratio created [13].

## 2. Materials and Methods

### 2.1. Social Return on Investment Methodology

The National Institute for Health and Care Excellence (NICE) advocates the use of both cost–benefit analysis (CBA) and cost–utility analysis (CUA) for evaluating public health interventions [14]. Social CBA is recommended in the HM Treasury Green Book for assessing the impact of interventions on wellbeing [15,16]. SROI is a pragmatic form of social CBA that uses quantitative and qualitative methods to value relevant costs and outcomes.

SROI methodology is outlined in the Cabinet Office Guide to social return on investment [17]. SROI takes a societal perspective and considers outcomes that are relevant and significant to stakeholders. SROI then assigns monetary values to these outcomes, which often do not have market prices. Examples of relevant outcomes in this EMD study are increased levels of mental wellbeing and self-efficacy experienced by clients. Using wellbeing valuation, the social value of relevant outcomes was then compared with the total costs to estimate an SROI ratio.

Wellbeing valuation offers a consistent and robust method for estimating the monetary value of outcomes that do not have market values. Wellbeing valuation can be applied using two social value calculators: the social value calculator derived from the Social Value Bank (SVB) and the mental health social value calculator derived from the short Warwick–Edinburgh Mental Wellbeing Scale (SWEMWBS).

In this study, the social value calculator was used to assign a monetary value to the outcome of increased self-efficacy, as measured by GSES scores, and the mental health social value calculator was used to assign a monetary value to mental wellbeing, as determined by SWEMWBS scores. Because the values in the social value calculator incorporate mental wellbeing, the two calculators were treated separately, with each generating its own SROI ratio [18] (Table 1).

The aim of this study was to establish how inputs (costs) were converted into outputs (numbers of clients) and subsequently into outcomes (improved mental wellbeing and self-efficacy). The social value generated by these outcomes was then estimated using a method similar to cost–benefit analysis, with a ratio comparing the cost per client with the social value generated per client. The SROI analysis was operationalised through the six stages outlined in the Guide to social return on investment [17]:Identifying stakeholders;Developing a theory of change;Calculating inputs;Evidencing and valuing outcomes;Establishing impact;Calculating the SROI ratio.

### 2.2. Identifying Stakeholders

The primary stakeholders were both the face-to-face and online clients, who directly experienced EMD coaching, and the National Health Service (NHS), which experienced a change in mental health service resource use by EMD clients. All participants in the EMD programme were sign posted to the service either from primary care or third sector organisations, with all 32 participants in the pilot study choosing to self-. At the time when the pilot study was conducted, all 32 participants had either experienced or were experiencing anxiety, stress, PTSD, OCD, or depression. This pilot study was carried out between May 2021 and March 2022, with data gathered from 15 previous face-to-face clients who completed ‘one-time-only’ questionnaires. In addition, data were collected from 17 new online clients during COVID-19 lockdowns; these clients completed baseline and follow-up questionnaires.

Participant outcome data and mental health service resource use data were collected from clients. Due to the scope of this study, data were not collected from other stakeholders who may have also benefited from EMD, such as family members of the participants.

Eligibility in this study included adults (over 18 years old) who were experiencing a physical, mental, or social issue that could benefit from EMD coaching. All clients required the ability to speak Welsh or English and the mental capacity to be able to reflect on their own wellbeing.

### 2.3. Theory of Change

A theory of change model was created to identify the expected changes experienced by participants. Often used in programme development and evaluation, theory of change models illustrate the links between the inputs, outputs, outcomes, and impact (Figure 2):

### 2.4. Calculating Inputs

The total costs for the EMD programmes included product development costs, consultancy costs, website costs, equipment and software costs, overhead costs, and staff costs. Additional start-up expenditures for product development and business consultancy were amortised over a period of 180 months.

#### 2.4.1. Product development costs

Product development costs included the writing and editing of the EMD programme as well as online personality profile testing.

#### 2.4.2. Consultancy costs

Consultancy costs included business development consulting, licensing development, marketing and sales consulting, and public speaking training.

#### 2.4.3. Website costs

Website costs included website maintenance, the website domain name, and the cost of a content editor. The content editor was responsible for the visual design of the website and social media content.

#### 2.4.4. Equipment and software costs

Equipment and software costs included the cost of a laptop, a mobile phone contract, an internet connection, Zoom, Calendly, and online cloud storage.

#### 2.4.5. Overhead costs

Overhead costs included ongoing operation costs such as insurance, accounting, and the cost of a home office space.

#### 2.4.6. Staff costs

Staff costs included the hourly rate for an EMD practitioner, which was based on the mean hourly rate of a U.K. lifestyle coach (GBP 45.78/h) [19,20], and an hourly rate of GBP 8.91 (the National Living Wage) for an administrative assistant working part-time (25 h per week) [21].

### 2.5. Evidencing and Valuing Outcomes

To accurately report improvements in mental wellbeing, this pilot SROI evaluation used a mixed-method approach to gather data. Pre-validated tools, namely the short Warwick–Edinburgh Mental Wellbeing Scale (SWEMWS) and the General Self-Efficacy Scale (GSES) [22,23], were used to gather information on psychological functioning and to ensure that a rigorous depiction of actual mental wellbeing was captured. In addition, participant interviews were conducted to gather views, experiences, perceptions, and attitudes concerning the EMD programme in order to understand what works for people and why. The main focus of this pilot EMD SROI evaluation consisted of using social cost–benefit analysis to take account of the social, environmental, and economic outcomes of the EMD social prescribing intervention as per the Social Value Act (2012) and the Wellbeing for Future Generations Act (2015). The secondary outcome of this pilot study was to estimate the varying levels of social value generated according to the means of delivery of the programme, which was either online or face-to-face, and to measure the actual associated social value generated.

#### 2.5.1. Questionnaires

The SROI evaluation included 15 previous face-to-face clients who completed a ‘one-time-only’ questionnaire and 17 online clients who completed baseline and follow-up questionnaires. The questionnaires captured demographic information, the reason for referral, baseline and follow-up health states, health service resource use, and additional questions about the client’s experience of EMD coaching. Questionnaire data were analysed to determine, for each outcome, the number of clients who improved, stayed the same, or worsened. Questionnaires included validated scales for assessing mental wellbeing and self-efficacy and an adapted client service receipt inventory (CSRI) form.

The short Warwick–Edinburgh Mental Wellbeing Scale (SWEMWBS) is a list of seven positively worded statements with five response categories that measure different aspects of positive mental health [24]. Overall scores can range from 7 to 35. The General Self-Efficacy Scale (GSES) is a 10-item self-reported measure of self-efficacy. It assesses the strength of an individual’s belief in their ability to respond to novel or difficult situations and to deal with any associated obstacles or setbacks [22]. Overall scores can range from 10 to 40.

#### 2.5.2. Client Service Receipt Inventory (CSRI)

An adapted CSRI form was used to record the number of mental-health-related visits that clients had with primary care health professionals (i.e., GPs and nurses) and with a community mental health team (i.e., clinical psychologists and mental health nurses). Clients reported the number of visits to health professionals for three months prior to and three months during EMD.

To ensure rigor in these research results, the stated preference technique of contingent valuation (CV) was incorporated into the post-evaluation questionnaires. The purpose of including CV questions was to understand clients’ choices and preferences, as well as the health benefits associated with participating in the EMD programme.

#### 2.5.3. Interviews

In addition to completing questionnaires, eight face-to-face clients and eight online clients attended an interview of approximately 30 min in length. Facilitated by a Bangor University researcher, the interview took place online without the presence of EMD coaches or support staff. The purpose of the interview was to further explore the clients’ experience of EMD coaching. Informed consent was obtained from clients prior to being interviewed. Interviews were audio recorded and transcribed.

#### 2.5.4. Wellbeing Valuation using the Social Value Calculator

Once the data were quantified, wellbeing valuation was applied to place a monetary value on the quantity of change. The social value calculator uses values from the HACT Social Value Bank (SVB), which includes approximately 120 methodologically consistent and robust social values. Often used in SROI and social CBA, these values provide a basic assessment of social value. In this study, a ‘high confidence’ value of GBP 13,080 was the monetary value assigned to an improvement in the GSES scale of 5 points or more. This was the value assigned to clients who improved from ‘low confidence’ to ‘high confidence’.

#### 2.5.5. Wellbeing Valuation using the Mental Health Social Value Calculator

Using the mental health social value calculator, baseline and follow-up SWEMWBS scores for each client were recorded, and values were assigned. Five steps were applied for calculating the social value using SWEMWBS [18] (Figure 3):

### 2.6. Establishing Impact

To avoid over-claiming using the HACT social value calculator, deadweight, attribution, and displacement were considered.

Deadweight reflects the possibility that a proportion of the outcomes would have happened anyway without the EMD programme. In this study, the follow-up questionnaire asked clients: *“How much of this change would have happened anyway (if you had not participated in the EMD coaching programme)?”*

Attribution acknowledges that a proportion of the outcomes could be attributable to factors other than the programme. In this study, the follow-up questionnaire asked clients: *“How much of this change was due to the EMD coaching programme?”*

Displacement considers whether participants had to give up any other activities that could have contributed to their wellbeing. In this study, the follow-up questionnaire asked participants: *“By participating in the EMD coaching programme over the last several months, how much have you had to give up other activities that benefitted your health and wellbeing?”*

To avoid over-claiming using the HACT mental health social value calculator, a 27% standard deadweight percentage for health outcomes was subtracted from SWEMWBS values, as recommended by the Housing and Communities Agency [25].

### 2.7. Calculating the SROI Ratio

Using the social value calculator and mental health social value calculator, wellbeing valuation generates SROI ratios that compare the social value of relevant outcomes with the total costs (Equation (1)).
(1)$$\mathrm{SROI}\;\mathrm{ratio}=\frac{\mathrm{Social}\;\mathrm{value}\;\mathrm{of}\;\mathrm{EMD}\;\mathrm{client}\;\mathrm{outcomes}}{\mathrm{Cost}\;\mathrm{of}\;\mathrm{delivering}\;\mathrm{EMD}\;\mathrm{programmes}}$$

## 3. Results

Analysis of questionnaire data indicated that both face-to-face and online clients were similar in terms of age, gender, and weekly household income. Differences were noted in relation to employment status, with face-to-face clients predominantly employed and online clients mainly unemployed. All clients indicated that the reason for participating in the EMD programme was that they were experiencing anxiety, stress, PTSD, or depression. Some clients had been sign posted from primary care. The online clients’ experience of self-enrolment in the EMD programme contrasted with that of the face-to-face clients, who were sign posted from primary care and self-referred to the EMD programme. The difference in SROI ratios between the face-to-face and online formats was also influenced by slightly different mean scores at follow-up for the SWEMWBS and GSES, as shown in Table 2. Reported mental wellbeing and self-confidence improvements for face-to-face and online clients are shown in Table 3 and Table 4.

The contingent valuation question asked clients from both cohorts the value they placed on the health benefit of participating in the EMD programme. These valuations ranged from GBP 0 to GBP 1200+. On average, the face-to-face clients indicated, on average, a willingness to pay (WTP) a sum of GBP 730 to participate in the EMD programme. The online clients indicated, on average, a WTP a sum of GBP 600 to participate in the EMD programme.

The WTP estimates indicated that clients were willing to pay for the health benefits of the EMD programme, which included improved mental wellbeing and self-confidence. In addition, the face-to-face clients indicated a WTP for participation in the EMD programme and its associated health and wellbeing improvements that was GBP 130 higher than online clients. The variance in WTP valuations suggests that face-to-face clients placed a higher value on a delivery approach that offered access to direct contact with the EMD programme compared with online clients.

### 3.1. Comparing Costs

Both face-to-face and online formats incurred similar costs. The main difference in costs was associated with the time it took to staff the two formats (Table 5). Typically, the EMD practitioner spent 18 h of 1:1 contact time with face-to-face clients compared with three hours of 1:1 contact with online clients. Online clients also received nine hours of group sessions, four of which were live group sessions; five hours consisted of online life reflection. Staffing and consultancy costs were the two largest cost categories for both EMD modalities. The total costs per client for the face-to-face and online EMD programmes were GBP 2,248 and GBP 1,430, respectively (Table 5).

### 3.2. Outcomes using the Social Value Calculator

To be able to quantify changes in self-efficacy (confidence), data were included only for participants who completed both baseline and follow-up questionnaires (n = 15 for face-to-face clients; n = 17 for online clients) (Table 6).

**Face-to-face participants:** When the number of participants who decreased in confidence by five points or more (n = 0) was subtracted from the number of clients who improved by five points or more (n = 15), the net increase was 15 participants. When 15 was multiplied by GBP 13,080, the total social value for high confidence among face-to-face clients was GBP 196,200 per year.

**Online participants:** When the number of participants who decreased in confidence by five points or more (n = 0) was subtracted from the number of participants who improved by five points or more on GSES (n = 10), the net increase was 10 participants. When 10 was multiplied by GBP 13,080, the total social value among online clients was GBP 130,800 per year.

**Social Impact:** When deadweight, attribution, and displacement were considered, follow-up questionnaire data indicated that the mean deadweight percentage was 10% for face-to-face clients and 19% for online clients. The attribution percentage was 83% for face-to-face clients and 60% for online clients. The displacement percentage was 8% for face-to-face clients and 10% for online clients. Therefore, the total social value for clients experiencing high confidence was GBP 8989 per client per year for face-to-face clients and GBP 3365 per client per year for online clients (Table 6).

### 3.3. Outcomes using the Mental Health Social Value Calculator

Using the five-step methodology for calculating social value using SWEMWBS, the social value was GBP 15,640 per client per year for face-to-face clients (Table 7) and GBP 4758 for online clients (Table 8).

### 3.4. Outcomes from the CSRI Questionnaire

The CSRI questionnaires completed by face-to-face and online clients measured health service resource use by comparing the number of mental-health-related visits to NHS professionals. Clients were asked about the number of mental-health-related visits for two different time periods, i.e., the three months preceding their lifestyle coaching programme and three months during their programme. The total annual cost saving was GBP 272 per face-to-face client and GBP 27 per online client (Table 9).

### 3.5. Outcomes (Non-Monetised) from Interviews

The qualitative data from interviews with face-to-face and online clients indicated perceived benefits attested to by participants in terms of both mental wellbeing and confidence (Figure 4).

### 3.6. Calculating the SROI Ratio

SROI ratios were calculated using the social value calculator and the mental health social value calculator (Table 10). When the total financial value per client was compared with the total cost per client, the SROI ratios ranged from GBP 4.12 to GBP 7.08 for every GBP 1 invested for face-to-face clients and from GBP 2.36 to GBP 3.34 for every GBP 1 invested for online clients (Table 10).

## 4. Discussion

Although the results of this evaluation indicate a positive social return on investment for both face-to-face and online EMD coaching, the SROI ratios were two to three times higher when lifestyle coaching was delivered via a face-to-face format. The significantly higher SROI ratios for face-to-face coaching may be due to the fact that it included five times more 1:1 coaching time than the online clients received. This suggests that the positive impact of the therapeutic relationship between coach and client was significant. The therapeutic relationship has been called the ‘foundation of mental health practice’, and a vast body of literature emphasises the importance of the therapeutic relationship in mental health [26].

The difference in SROI ratios between face-to-face and online formats was influenced by significantly different mean scores at baseline for both the SWEMWBS and GSES. Although there was homogeneity among the face-to-face and online cohorts in terms of age, gender, reason for enrolment, and mean weekly household income, the face-to-face clients were predominantly unemployed and referred from GPs. Alternatively, the online clients were mostly employed and self-referred. Research indicates that unemployed individuals are more likely to report lower mental health than people in employment [27].

The difference in SROI ratios between face-to-face and online formats was also influenced by slightly different mean scores at follow-up for both the SWEMWBS and GSES. Face-to-face clients reported slightly higher mean scores at follow-up, which may have been due to more 1:1 contact and/or to face-to-face clients having more time to use EMD tools and knowledge. Face-to-face clients completed the ‘one-time-only’ questionnaire many months (11–44 months) after completing EMD, which may have enabled them time to internalise the EMD tools. The online clients, on the other hand, completed the questionnaire within a week of finishing the EMD programme.

### 4.1. Strengths

Although previous studies have investigated the effectiveness of lifestyle coaching for improving mental wellbeing [28,29], this is the first study to undertake an SROI evaluation of lifestyle coaching. This study also applied a mixed-method approach that used both qualitative and quantitative data. The quantitative data were strengthened with two valid and reliable outcome measures, i.e., the SWEMWBS and the GSES. Furthermore, the SROI ratios were generated from two wellbeing valuation sources: the HACT social value calculator and the HACT mental health social value calculator. Both calculators are derived from wellbeing valuation, a consistent and robust method recommended in HM Treasury’s Green Book (2018) for measuring social CBA.

Finally, EMD coaching aligns with the NHS Five Year Forward View, which encourages the development of new empowerment-based interventions to supplement existing mental health programmes. By estimating the social return on investment of EMD, this study provides important data to support evidence-based decision making in primary care settings.

### 4.2. Limitations

This is a pilot study of the EMD coaching programme; therefore, the results should be treated with caution. Our recommendation is that future evaluation of the EMD programme should be expanded to a feasibility trial to further demonstrate the effectiveness of the coaching programme in supporting improvements in mental wellbeing. It is acknowledged that this evaluation of the pilot EMD programme did not employ an experimental study design; therefore, the study did not include a control group, and this could be perceived as a limitation. However, the purpose of this evaluation was to consider the effectiveness of the programme in enhancing mental wellbeing and to generate and assess an associated social cost–benefit analysis. The secondary outcomes of this pilot study were the valuation of the two EMD programme delivery approaches, i.e., face-to-face and online, and the estimation of the associated social value ratio created.

It is acknowledged that the sample size was small; however, evidence indicates that the mixed-method approach applied in SROI is not reliant on large sample sizes because the analysis is operationalised through the six stages [17]; this approach can determine the social value of a type of intervention, rather than a specific intervention such as the EMD programme in this research [29]. In addition, the online version of the EMD programme was delivered during the COVID-19 pandemic, which made recruitment challenging and impacted participant numbers.

Although this study used valid and reliable questionnaires, it was only possible to collect data retrospectively from face-to-face clients due to COVID-19. It is likely that recall bias may have affected the accuracy of the baseline scores in the ‘one-time-only’ questionnaire completed by face-to-face clients, who may not have correctly recalled their actual mental wellbeing and confidence at the start of their EMD programme.

Selection bias may also have affected the findings with respect to face-to-face clients. Approximately 21% of previous face-to-face clients (n = 15/70) completed the ‘one-time-only’ questionnaire. It could be that the 15 EMD clients who responded were those who benefitted the most from the programme. Finally, this study lacked a control group, which means that there is a possibility that client improvement in mental wellbeing and self-efficacy could have been due to other factors. However, this limitation was mitigated by the 27% deadweight percentage (the proportion of outcomes that would have taken place anyway without the EMD programme) applied when using the HACT mental health social value calculator and self-reporting percentages for deadweight, attribution, and displacement when using the HACT social value calculator.

## 5. Conclusions

The results indicated that the face-to-face EMD format generated positive SROI ratios ranging from GBP 4.12 to GBP 7.08 for every GBP 1 invested. SROI ratios for the online format ranged from GBP 2.37 to GBP 3.35. Quantitative and qualitative data indicated that the EMD coaching programme intervention facilitated improved mental wellbeing and self-efficacy for clients.

The NHS Long Term Plan (2019–2024) stated a renewed commitment to improving and widening access to care for people needing mental health support [30]. New standards established by NHS England and NHS Improvement state that people seeking mental health support in the community should get help within four weeks, while those with an urgent mental health need should be seen by a community crisis team within 24 h [31].

The recent COVID-19 crisis demonstrated the importance of making digital mental health interventions a routine part of care [32]. Research indicates that online self-help interventions for individuals experiencing psychological distress can be effective for improving emotional regulation skills and resilience [33].

Further research and development are needed to ensure that positive therapeutic relationships are created and maintained when using online mental health interventions, such as EMD lifestyle coaching [34]. With the continued existence of long waiting lists for people with mental health challenges, face-to-face and online lifestyle coaching may become an important service across the statutory, private, and third sectors to meet the growing demand for mental health support.

## Figures and Tables

**Figure 1 ijerph-19-10658-f001:**
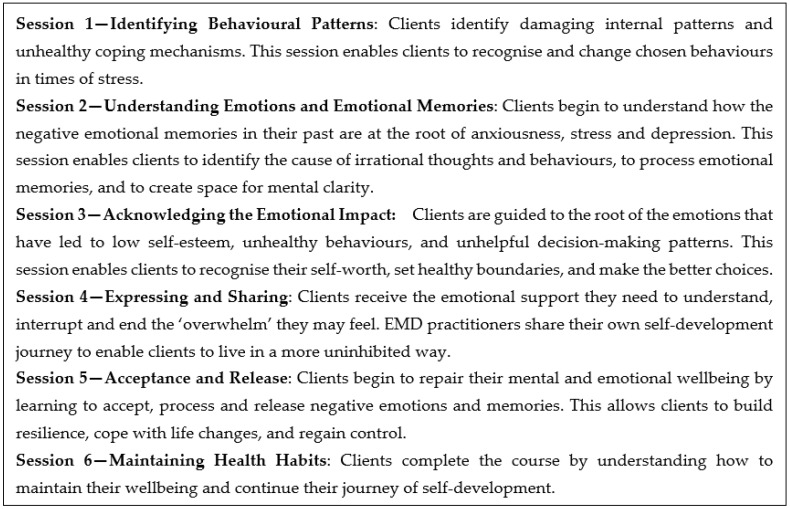
EMD lifestyle coaching programme.

**Figure 2 ijerph-19-10658-f002:**
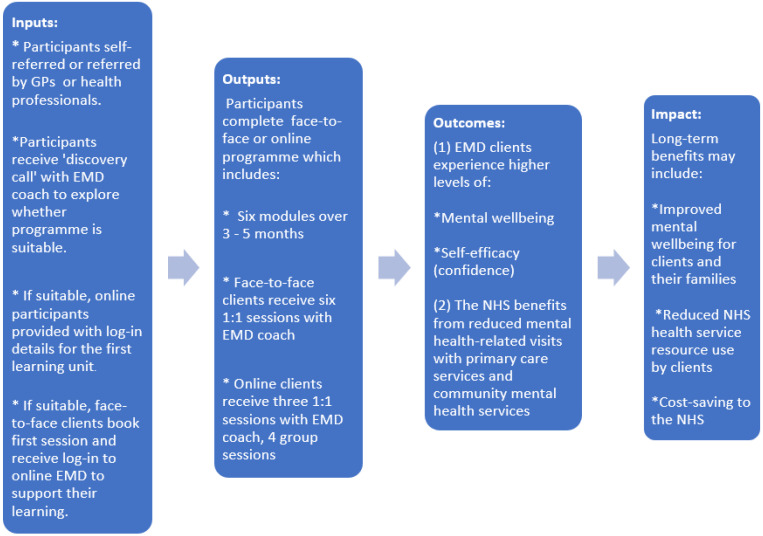
Theory of change model.

**Figure 3 ijerph-19-10658-f003:**
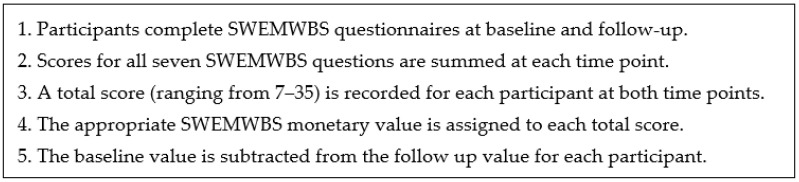
Calculating social value using SWEMWBS.

**Figure 4 ijerph-19-10658-f004:**
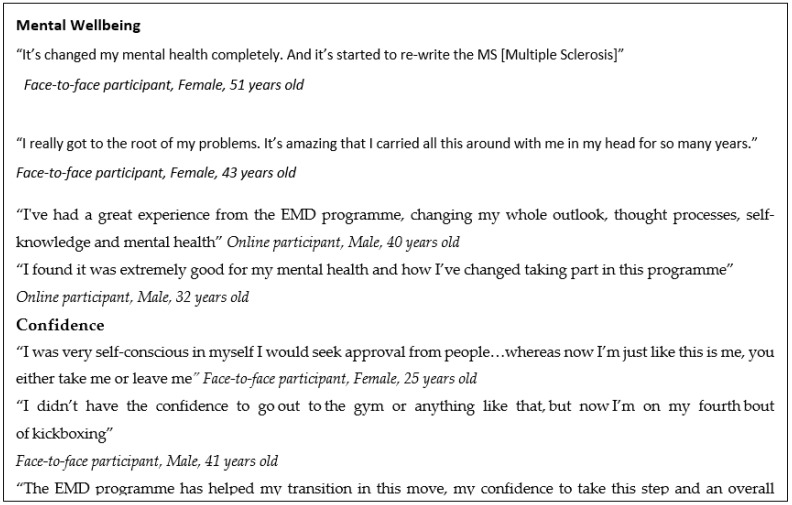
Selected quotes from face-to-face and online participants.

**Table 1 ijerph-19-10658-t001:** Wellbeing valuation methods.

Outcome	Outcome Measure	Wellbeing Valuation Method
Mental wellbeing	SWEMWBS	Mental health social value calculator v.1.0
Self-efficacy	GSES	Social value calculator v.4.0

**Table 2 ijerph-19-10658-t002:** **Overview of EMD clients.**

	Face-To-Face Clients	Online Clients
**Mean age**	43 years old	44 years old
**Gender**	73% female, 27% male	65% female, 35% male
**Ethnic origin**	100% White British	100% White British
**Main reason for enrolment**	67% enrolled citing depression	47% enrolled citing depression
**Weekly household income**	GBP 325	GBP 323
**Willingness to pay for EMD**	GBP 730	GBP 600
**Mean SWEMWBS score at baseline**	13	20
**Mean SWEMWBS score at follow-up**	28	26
**Mean GSES score at baseline**	16	26
**Mean GSES score at follow-up**	35	31

**Table 3 ijerph-19-10658-t003:** Reported improvements for face-to-face and online clients for Outcome 1: Mental wellbeing.

	Face-To-Face Clients	Online Clients
Reported improvement of 1 point or more	100% (15/15)	88% (15/17)
Reported improvement of 5 points or more	100% (15/15)	65% (11/17)
Reported improvement of 10 points or more	93% (14/15)	29% (5/17)

**Table 4 ijerph-19-10658-t004:** Reported improvements for face-to-face and online clients for Outcome 2: Self-confidence.

	Face-To-Face Clients	Online Clients
Reported improvement of 1 point or more	100% (15/15)	82% (14/17)
Reported improvement of 5 points or more	100% (15/15)	59% (10/17)
Reported improvement of 10 points or more	93% (14/15)	18% (3/17)

**Table 5 ijerph-19-10658-t005:** Annual costs to deliver the EMD programme.

Cost Category	Face-To-Face EMD Programme	Online EMD Programme
**Product development costs** 150 h writing and editing 31-unit programme 62 h updating 31 units of online programme Online personality profiling training	**GBP 488 with 180-month amortisation** GBP 458 with 180-month amortisation n/a GBP 30 with 180-month amortisation	**GBP 677 with 180-month amortisation** GBP 458 with 180-month amortisation GBP 189 with 180-month amortisation GBP 30 with 180-month amortisation
**Consultancy costs** Business development consultant Licensing development consultant Public speaking consultant Marketing and sales consultants	**GBP 1179 with 180-month amortisation** GBP 617 with 180-month amortisation GBP 33 with 180-month amortisation GBP 240 with 180-month amortisation GBP 289 with 180-month amortisation	**GBP 1179 with 180-month amortisation** GBP 617 with 180-month amortisation GBP 33 with 180-month amortisation GBP 240 with 180-month amortisation GBP 289 with 180-month amortisation
**Website costs** Website programmer Website domain name Website content editor	**GBP 5736** GBP 4080 GBP 256 GBP 1400	**GBP 5736** GBP 4080 GBP 256 GBP 1400
**Equipment and software costs** Cost of a laptop Mobile phone Internet Cloud storage Video-conferencing license Online booking platform license	**GBP 904** GBP 300 GBP 192 GBP 120 GBP 30 GBP 117 GBP 145	**GBP 904** GBP 300 GBP 192 GBP 120 GBP 30 GBP 117 GBP 145
**Overhead costs** Insurance costs Accounting costs Cost of EMD office	**GBP 1477** GBP 84 GBP 1158 GBP 235	**GBP 1477** GBP 84 GBP 1158 GBP 235
**Staffing costs** **(1)** **EMD practitioner costs** 1:1 contact programme delivery Life reflection sessions (**online**) Group sessions (**online**) **(2)** **Part-time admin assistant**	**GBP 23,944** GBP 12,361 n/a n/a GBP 11,583	**GBP 14,330** GBP 2335 GBP 183 GBP 229 GBP 11,583
**Total annual cost with 180-month amortisation for start-up product development costs and consultancy costs**	**GBP 33,728**	**GBP 24,303**
**Total cost per client per year**	**GBP 2248 (n = 15)**	**GBP 1430 (n = 17)**

**Table 6 ijerph-19-10658-t006:** Quantity of outcomes and social value for high confidence (self-efficacy).

Outcome: Confidence	Net Quantity	Financial Value (per Annum)	Total Social Value (per Annum)	Deadweight	Attribution	Displacement	Total Social Value (per Annum)	Social Value per Client
**Face-to-face** **(n = 15)**	15/15	GBP 13,080	GBP 196,200	10% (×0.9)	17% (×0.83)	8% (×0.92)	GBP 134,836	GBP 8989
**Online** **(n = 17)**	10/17	GBP 13,080	GBP 130,800	19% (×0.81)	40% (×0.60)	10% (×0.90)	GBP 57,212	GBP 3365

**Table 7 ijerph-19-10658-t007:** Social value for face-to-face clients using the mental health social value calculator.

ID	Baseline (T1)	T1 Value	Follow-Up (T2)	T2 Value	Difference (T2-T1)	After Deadweight (27%)
1	13	0	35	GBP 26,793	GBP 26,793	GBP 19,559
2	11	0	33	GBP 26,175	GBP 26,175	GBP 19,108
3	19	GBP 17,561	33	GBP 26,175	GBP 8,614	GBP 6288
4	7	0	35	GBP 26,793	GBP 26,793	GBP 19,559
5	10	0	20	GBP 17,561	GBP 17,561	GBP 12,820
6	14	0	31	GBP 25,856	GBP 25,856	GBP 18,875
7	13	0	28	GBP 24,877	GBP 24,877	GBP 18,160
8	12	0	29	GBP 25,480	GBP 25,480	GBP 18,600
9	12	0	30	GBP 25,480	GBP 25,480	GBP 18,600
10	9	0	24	GBP 22,944	GBP 22,944	GBP 16,749
11	12	0	29	GBP 25,480	GBP 25,480	GBP 18,600
12	8	0	27	GBP 24,877	GBP 24,877	GBP 18,160
13	16	GBP 9639	27	GBP 24,877	GBP 15,238	GBP 11,124
14	17	GBP 12,255	25	GBP 24,225	GBP 11,970	GBP 8738
15	18	GBP 12,255	29	GBP 25,480	GBP 13,225	GBP 9654
**Total**		**GBP 51,710.00**		**GBP 373,073.00**	**GBP 321,363.00**	**GBP 234,595**
**Total social value per client (n = 15)**						**GBP 15,640**

**Table 8 ijerph-19-10658-t008:** Social value for online clients using the mental health social value calculator.

ID	Baseline (T1)	T1 Value	Follow-Up (T2)	T2 Value	Difference (T2-T1)	After Deadweight (27%)
1	26	GBP 24,225.00	29	GBP 25,480.00	GBP 1,255.00	GBP 916
2	8	GBP 0	22	GBP 21,049.00	GBP 21,049.00	GBP 15,365
3	18	GBP 12,255.00	15	GBP 9,639.00	−GBP 2616.00	−GBP 1910
4	17	GBP 12,255.00	28	GBP 24,877.00	GBP 12,622.00	GBP 9214
5	21	GBP 21,049.00	28	GBP 24,877.00	GBP 3828.00	GBP 2794
6	20	GBP 17,561.00	26	GBP 24,225.00	GBP 6664.00	GBP 4865
7	21	GBP 21,049.00	28	GBP 24,877.00	GBP 3828.00	GBP 2794
8	13	GBP 0	28	GBP 24,877.00	GBP 24,877.00	GBP 18,160
9	18	GBP 12,255.00	24	GBP 22,944.00	GBP 10,689.00	GBP 7803
10	23	GBP 22,944.00	34	GBP 26,175.00	GBP 3231.00	GBP 2359
11	24	GBP 22,944.00	26	GBP 24,225.00	GBP 1281.00	GBP 935
12	22	GBP 21,049.00	25	GBP 24,225.00	GBP 3176.00	GBP 2318
13	21	GBP 21,049.00	28	GBP 24,877.00	GBP 3828.00	GBP 2794
14	28	GBP 24,877.00	27	GBP 24,877.00	GBP 0	GBP 0
15	18	GBP 12,255.00	25	GBP 24,225.00	GBP 11,970.00	GBP 8738
16	21	GBP 21,049.00	22	GBP 21,049.00	GBP 0	GBP 0
17	22	GBP 21,049.00	34	GBP 26,175.00	GBP 5126.00	GBP 3742
**Total**		**GBP 287,865.00**		**GBP 398,673.00**	**GBP 110,808.00**	**GBP 80,890**
**Total social value per client (n = 17)**						**GBP 4758**

**Table 9 ijerph-19-10658-t009:** Health service resource use for face-to-face and online clients.

Type of client	3 Months before Programme	3 Months during Programme	Difference in Visits	Cost per Visit	Cost Saving per 3 Months	Cost Saving per 12 Months
**Face-to-face clients (n = 15)**						
GP visits	19	9	10	GBP 39/visit ^1^	GBP 390	GBP 1,560
Nurse	2	0	2	GBP 44/visit ^1^	GBP 88	GBP 352
Psychologist	9	0	9	GBP 58/visit ^1^	GBP 522	GBP 2088
Mental health nurse	1	0	1	GBP 21/visit ^1^	GBP 21	GBP 84
**Total cost saving**					**GBP 1021**	**GBP 4084**
**Total cost saving per face-to-face client**						**GBP 272**
**Online clients (n = 17)**						
GP visits	5	1	4	GBP 39/visit ^1^	GBP 156	GBP 624
Nurse	1	1	0	GBP 44/visit ^1^	GBP 0	GBP 0
Psychologist	1	1	0	GBP 58/visit ^1^	GBP 0	GBP 0
Mental health nurse	1	3	-2	GBP 21/visit ^1^	−GBP 42	−GBP 168
**Total cost saving**					**GBP 114**	**GBP 456**
**Total cost saving per online client**						**GBP 27**

^1^ PSSRU, 2021.

**Table 10 ijerph-19-10658-t010:** SROI ratios using the social value calculator and the mental health social value calculator.

	SROI Ratio (Social Value Calculator)	SROI Ratio (Mental Health Social Value Calculator)
Total social value per face-to-face client	GBP 8989	GBP 15,640
NHS cost savings per face-to-face client	GBP 272	GBP 272
Total financial value per face-to-face client	GBP 9261	GBP 15,912
Total cost per face-to-face client	GBP 2248	GBP 2248
**SROI ratio for face-to-face clients**	**GBP 4.12: GBP 1**	**GBP 7.08: GBP 1**
Total social value per online client	GBP 3365	GBP 4758
NHS cost savings per online client	GBP 27	GBP 27
Total financial value per online client	GBP 3392	GBP 4785
Total cost per online client	GBP 1430	GBP 1430
**SROI ratio for online clients**	**GBP 2.37: GBP 1**	**GBP 3.35: GBP 1**

## Data Availability

The final dataset will only be available to the study investigators and the advisory team. Informed consent was obtained from all subjects involved in the study.

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
