# Peer review of "A Social Return on Investment Evaluation of the Pilot Social Prescribing EmotionMind Dynamic Coaching Programme to Improve Mental Wellbeing and Self-Confidence"

_ijerph, 2022, doi:10.3390/ijerph191710658_

Round 1

Reviewer 1 Report

The manuscript entitled “Social return on investment of face-to-face versus online lifestyle coaching to improve mental wellbeing” describes a economical analysis of two forms of life style coaching. Although the possibility to use non-clinical approaches in order to improve a delivery of mental health care is justified, the design and analysis of the examined interventions have several limitations.

#1. A study lacks a control group. Thus, the obtained result indicate only differences between two forms of delivery of the same treatment, but only supposed efficiency. The lack of a control group (with typical treatment or other specified forms of treatment) are the main limitation of the study.

#2. The SOI analysis could be useful. However, I would like to see repeated ANOVAs or t test for dependent means in order to demonstrate psychologically interpretable differences between two measurements.

#3. The more detailed description of groups should be provided, e.g. in terms of mental health. Were randomization implemented? Did citing depression was analyzed as a moderator? How about the number of participants? Did the Authors conducted a priori power analysis?

#4. The measures which relies on the subjective opinion that a given method was responsible for a change in psychological functioning are relatively weak indicators.

Author Response

Please see the attachemnt.

Reviewer 2 Report

The study presents important findings regarding the positive social value ratios of face-to-face and online lifestyle coaching. Please provide additional information regarding the following aspects:

- The sample size is relatively small. Please provide information regarding the power of the study. 

- The main reason for study enrolment is depression in 67% of face-to-face clients and 47% in online clients. This raises questions regarding the degree to which the participants in the two group are matched to each other. Please state whether the rate of depression differs between the groups.

Round 2

Reviewer 1 Report

The Authors did not revised their manuscript substantially. I failed to find in their response a clear justification of why the SROI analyses which they conducted and which is important was not support by addotional psychological analysis. I failed also to find the answers to the other reviewer's questions.

My opinion is that when the Authors state: "The purpose of the pilot EMD SROI evaluation was to appraise the effectiveness of 102 the programme in enhancing mental wellbeing", they mean that they will analyse the differences at least between mental wellbeing befire and after an attendance to EMD programme. It is not an argument that the study was a pilot study. Also the pilot study has to be conducted with an appripriate caution. If the Authors would like only to conduct SOR analysis - it is ok, but they should not state that they are studying the effectiveness of EMD programme for wellbeing.

Thus, in my opinion, the analyses should at least be supported by  descriptive statistics on mental well-being at two time point during the study (fot both modalities of EMD application). The readers could make their own calculation using this data in order to understand the therapeutic usefellness of the EMD. 
